# The Association Pattern between Ambient Temperature Change and Leukocyte Counts

**DOI:** 10.3390/ijerph18136971

**Published:** 2021-06-29

**Authors:** Shih-Chiang Hung, Chen-Cheng Yang, Chu-Feng Liu, Chia-Te Kung, Wen-Huei Lee, Chi-Kung Ho, Hung-Yi Chuang, Hsin-Su Yu

**Affiliations:** 1Department of Public Health, College of Health Sciences, Kaohsiung Medical University, Kaohsiung 807, Taiwan; hsc0901@cgmh.org.tw (S.-C.H.); kmco6849@ms14.hinet.net (C.-K.H.); 2Department of Emergency Medicine, Kaohsiung Chang Gung Memorial Hospital and Chang Gung University College of Medicine, Kaohsiung 83301, Taiwan; leou4503@cgmh.org.tw (C.-F.L.); g00308@cgmh.org.tw (C.-T.K.); malee4950@cgmh.org.tw (W.-H.L.); 3Department of Occupational Medicine and Family Medicine, Kaohsiung Municipal Siaogang Hospital and Kaohsiung Medical University, Kaohsiung 807, Taiwan; abcmacoto@gmail.com; 4Department of Occupational and Environmental Medicine, Kaohsiung Medical University Hospital, Kaohsiung 807, Taiwan; 5Department of Public Health and Environmental Medicine, Research Center for Environmental Medicine, College of Medicine, Kaohsiung Medical University, Kaohsiung 807, Taiwan; 6College of Medicine, Kaohsiung Medical University, Kaohsiung 807, Taiwan; dermyu@kmu.edu.tw

**Keywords:** ambient temperature, white blood cell count, generalized additive model

## Abstract

Ambient temperature change is one of the risk factors of human health. Moreover, links between white blood cell counts (WBC) and diseases have been revealed in the literature. Still, we do not know of any association between ambient temperature change and WBC counts. The aim of our study is to investigate the relationship between ambient temperature change and WBC counts. We conducted this two-year population-based observational study in Kaohsiung city, recruiting voluntary community participants. Total WBC and differential counts, demographic data and health hazard habits were collected and matched with the meteorological data of air-quality monitoring stations with participants’ study dates and addresses. Generalized additive models (GAM) with penalized smoothing spline functions were performed for the trend of temperature changes and WBC counts. There were 9278 participants (45.3% male, aged 54.3 ± 5.9 years-old) included in analysis. Compared with stable weather conditions, the WBC counts were statistically higher when the one-day lag temperature changed over 2 degrees Celsius, regardless of whether colder or hotter. We found a V-shaped pattern association between WBC counts and temperature changes in GAM. The ambient temperature change was associated with WBC counts, and might imply an impact on systematic inflammation response.

## 1. Introduction

Ambient temperature change is one of the risk factors of human health. Previous studies have revealed the links between ambient temperature change and diseases, in both cold and hot weather [1,2,3,4]. The exposure to ambient temperature is one of the important determinants of disease burden [5], mortality and morbidity [6,7,8,9,10,11,12,13,14,15]. Moreover, some studies find the relationship between ambient temperature and morbidity is not linear, but a U-, V- or J-shaped pattern [16,17,18,19]. Therefore, some studies have found optimal temperatures corresponding to minimum mortality [20,21,22]. However, less is known about the mechanism explaining the association between air temperature and mortality or morbidity.

Peripheral white blood cell (WBC) count is an easy-accessed marker, representing the severity of systematic response to inflammation or stress. Meanwhile, we note many previous studies have reported the associations between peripheral WBC count and incidence of disease, morbidity and mortality, such as atrial fibrillation [23], stroke [24,25,26], myocardium infarction [27,28], respiratory disease [29,30,31,32,33], acute kidney disease [34], diabetes mellitus [35,36,37,38], etc. However, few studies explore the relationship of air temperature and peripheral WBC count.

Despite the fact that ambient temperature impacts health and peripheral WBC counts are associated with disease outcomes, studies exploring the relationship between ambient temperature change and WBC counts, and the mechanism of how ambient temperature variations influence human health are few [39,40]. Inflammation response and cascading reactions seem to be one of postulated mechanisms [41].

Therefore, we suppose that there might be a plausible link between environmental air temperature change and WBC counts. The goal of our study is to explore if any short-term association between ambient air temperature change and peripheral WBC counts exists.

## 2. Materials and Methods

This was a population-based observational study, conducted in 2003 and 2004, and approved by the Institutional Review Board of Kaohsiung Medical University Hospital (KMUH-IRB-990206). The study purpose and general method had been explained to each participant, and gotten the consent from each individual one. Kaohsiung Municipal Hsiaokang Hospital administered the clinical and laboratory examinations for study subjects and funds were sponsored by the Health Bureau of Kaohsiung Municipality. Adult residents of Kaohsiung City, who were stratifiedly sampled by proportion of population, were invited to participate in this health survey program by letters and telephone calls. Participants were volunteers who responded to the invitations. Demographic data including habitual behaviors were collected during the visit by questionnaire. Participants with pregnancy, current malignancy diseases, history of auto-immune diseases, status of infectious disease (such as common cold) or fever on the exam day, were excluded. Blood cell counts were analyzed by Sysmex XE-2100 hematology automated analyzer in the Hsiaokang Municipal Hospital laboratory immediately after blood drawing.

There were 43 meteorological stations set up by the Taiwan Central Weather Bureau and Government since 1993 (https://www.cwb.gov.tw/V8/C/W/OBS_County.html?ID=64, accessed on 7 April 2021). Daily meteorological data including air temperature (T_air_), dew point temperature (T_dewpt_), sea level pressure, total cloud cover, wind speed and wind direction were collected for the entire study period.

Apparent temperature (T_app_) based on air and dew point temperature, calculated using the following formula [42,43] (Equation (1)):T_app_ = −2.653 + 0.994 (T_air_) + 0.0153 (T_dewpt_)^2^(1)

We collected the hourly measurements of ambient temperature and dew point temperature from meteorological stations, and calculated the hourly apparent temperature based on each hourly measurement. Then we calculate the daily mean value based on hourly apparent temperature measurements. Finally, we tested the changes of apparent temperature with the previous day (Equation (2)):ΔT = T_app_ − T_app(i−1)_, (2)
where i is health exam day.

### Statistics

Weather data were merged to individual blood exam data by the address reported for statistical analysis and by the date of blood examination. For each person, we selected the daily average temperature data from the meteorological station nearest to his/her residence at the blood examination day as “examination day” (ED, or lag 0). Temperature data of the seven days before blood examination were identified as lag 1, lag 2, lag 3, lag 4, lag 5, lag 6, and lag 7 prospectively. Microsoft Office Excel (Microsoft, Redmond, DC, USA) and SPSS for Windows (IBM Corp., Armonk, NY, USA), and R language (R Core Team, Vienna, Austria) were used for descriptive analyses; one-way ANOVA, correlation coefficients, and multiple linear regressions were employed for inference analyses. Type I error was set as 0.05 with two-tailed. We tested the changes of apparent temperature with the previous day larger than 2 degrees Celsius (ΔT > 2 °C, ΔT < −2 °C).

In addition, assuming a U- or V-shaped pattern association between WBC counts and temperature changes from the literature review [16,17,18,19], generalized additive models (GAM) with penalized smoothing spline function were performed for the trend of temperature changes and dependent variables (WBC, neutrophil, monocyte, basophil, eosinophil, and lymphocyte). Briefly, a GAM with a linear predictor involving a sum of smooth functions of covariates [44] (Equation (3)):g(E(Y)) = b_0_ + f(X_1_) + f(X_2_) + ... + f(X_n_)(3)
where f is a smoothing function. In our analysis, we fit GAM models (Equation (4)):g(E(WBC)) = s(dT_app_) + β_2_ × sex + β_3_ × age + β_4_ × tobacco + β_5_ × alcohol(4)
where ‘WBC’ was total white blood cell counts; ‘dT_app_’ meant the difference of apparent temperature with the previous day, and ‘s’ was a spline function. Sex, age, alcohol consumption (yes, ≥3 times per week), and tobacco consumption (yes or no) were not fit spline function.

For individual white cell counts, i.e., neutrophil, monocyte, basophil, eosinophil, and lymphocyte, we fit the above model using individual white cell accounts instead of WBC as the dependent variable to test whether individual white cell count would be sensitive to temperature changes.

## 3. Results

Kaohsiung City is located in the southwestern Taiwan Island, and next to the Taiwan Strait. The climate is tropical monsoon type (Group Am by Köppen–Geiger climate classification). In 2003, the annual daily average ambient temperature was 24.7 °C, annual daily average humidity was 77.7%. Initially, there were 10,140 volunteers recruited. After matching with the ambient temperature records measured by meteorological monitoring stations located at the same districts with volunteers’ addresses. There were 9278 records included in the final analysis, because 862 records could not be matched with available temperature data. The steps of data collection are briefed in Figure 1. Among the study participants, 45.3% are male, the average age and body mass index were 54.3 ± 5.9 years-old and 24.7 ± 3.8 kg/m^2^, respectively. The prevalence of unhealth habits current smoking, alcohol consumption and betel quid use were 15.7%, 14.9% and 2.9%, respectively. There were 331 blood examining days. We divided them into four subgroups by the difference in one-day lag temperatures (ΔT, difference of apparent temperature between two consecutive days), ΔT > 2 °C, 2 °C ≥ ΔT ≥ 0 °C, −2 °C ≤ ΔT < 0 °C, and ΔT < −2°C. More information is detailed in Table 1.

Table 2 show the counts and percentages of total and differential WBCs. They were also grouped by difference in one-day lag temperature (ΔT) as mentioned above. The counts of total WBC, neutrophil, monocyte, and basophil were different statistically between groups. The more absolute difference of one-day lag temperature, the higher counts of total WBC, neutrophile, monocyte and basophil. We also found that no matter whether the weather was getting hotter or cooler, the change in total WBC counts were significant statistically. This is revealed in Figure 2.

Figure 3a shows the scatter plot of total WBC counts by one-day lag temperature change. A trend line is also attached. From the trend line, we can vaguely see that the greater the one-day lag temperature change, whether hotter or cooler, the greater the total WBC counts. With the GAM procedure, the enhanced scatter plot statistically reveals a V-shaped pattern. This is shown in Figure 3b.

Figure 4 shows the scatter plot graphs of counts of WBC differential types. Trend lines of the GAM procedure are also present. In all types except for lymphocyte, the *p*-values of the GAM analysis reach statistically significance, i.e., in neutrophil, monocyte, eosinophil and basophil.

## 4. Discussion

This was a population-based observational study attempting to find the exposure–response association between ambient temperature change and peripheral WBC counts, after adjusting for sex, age, and habits of smoking and alcohol consumption. The results of our research revealed that there is a V-shaped pattern association between ambient temperature change and WBC counts. The higher the apparent temperature difference between two consecutive days (no matter hotter or cooler) was associated with an increase in the WBC counts.

From a literature review, we found that some previous studies have reported a non-linear relationship, such as U-, V- or J- patterns [45,46,47], between air temperature and mortality. Our research finding, i.e., a V-shaped pattern association between apparent temperature variance and WBC count, is consistent with air temperature–mortality relationships of previous studies. This may imply that the factor of peripheral WBC count has a role in the temperature effect on disease. Studies have also found optimal temperatures, at which the mortality is minimum. However, the optimal temperature is varied across studies by different locations [45,48]. This might be owing to the adaption of local residents to their local climates. Therefore, the method of our research seems to be more reasonable by using relative change of ambient temperature as an exposure measurement.

The relationship between WBC count and disease has also been reported across research. Reports of associations between WBC count and the risk and outcomes of diseases such as stroke, cardiovascular disease, respiratory disease, diabetic mellitus and metabolic syndrome, are numerous.

In our research, the slopes of the bilateral arms of the V-shaped curve are different. The hot-side (right) arm is steeper than the cold-side (left) arm. This is also consistent with previous studies [9]. Bunker et al., investigated the cause-specific mortality and morbidity in the elderly, and found that for climate-sensitive non-infectious diseases, such as cardiovascular, respiratory and cerebrovascular disease, the air temperature hazard impact of the mortalities was more obvious on the hot-side. The mortality increase for every 1 °C temperature change (rise vs. reduction) was 3.44% (95% CI 3.10–3.78) vs. 1.66% (95% CI 1.19–2.14) for cardiovascular, 3.60% (95% CI 3.18–4.02) vs. 2.90% (95% CI 1.84–3.97) for respiratory, and 1.40% (95% CI 0.06–2.75) vs. 1.21% (95% CI 0.66–1.77) for cerebrovascular disease.

There are still some limitations of this study. First, the ambient temperatures we measured were derived from the data of fixed outdoor monitoring stations, not from personal mobile devices. This would result in systematic measurement errors. However, these errors should not alter the finding of the existence of the WBC–temperature association. Second, WBC count can be influenced by short-term stress or inflammation resulting from personal habits and behaviors. The characters and quantities of stress and inflammation could not be well designed, checked and calculated. However, we still tried adjusting for age, health-risk behaviors (smoking, alcohol consumption), and so stress related errors might be partially corrected. Third, this research was conducted in small local area (four districts, total area 92 square kilometers) and the study population was relative homogeneous. This will cast a restriction on external validity. However, the local area means the variation of exposure of air temperature of study population is not wide. Fifth, all the participants were voluntary, and they could have a health check-up. The healthy worker effect might be possible, for these participants would more care about their physical condition and have a relative lower prevalence of hazardous health habits. For example, in 2002, the general adult smoking rate was 27% [49] but the rate was only 15.7% among our study participants. Sixth, in addition to apparent temperature, there are several thermal indexes to quantify the thermal stress on the human body, such as the wet-bulb globe temperature (WBGT) and the physiological equivalent temperature (PET) which might be superior to apparent temperature for the representative of thermal physical stress [50,51]. Further research with these thermal indexes should be considered.

From this study, we found an association between WBC counts and one-day lag temperature difference. The WBC counts would be raised if the difference of apparent temperature between two consecutive days was over 2 °C. This might imply greater inflammation/stress response and diseases would be provoked by weather temperature changes. The government might consider making early warnings to people if dramatic apparent temperature changes are predicted for the next day, and making healthcare systems prepared.

## 5. Conclusions

We found a V-shaped pattern association between WBC counts and temperature changes in GAM. The ambient temperature change, no matter higher or lower, was associated with increased WBC counts. Only lymphocyte was not significant. this might imply an impact on systematic inflammation response.

## Figures and Tables

**Figure 1 ijerph-18-06971-f001:**
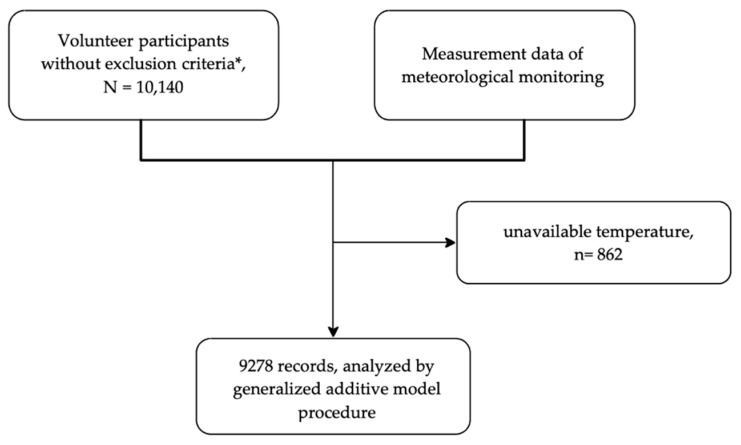
Brief of including records of participants. * exclusion criteria: pregnancy, current malignancy diseases, history of auto-immune diseases, status of infectious disease or fever.

**Figure 2 ijerph-18-06971-f002:**
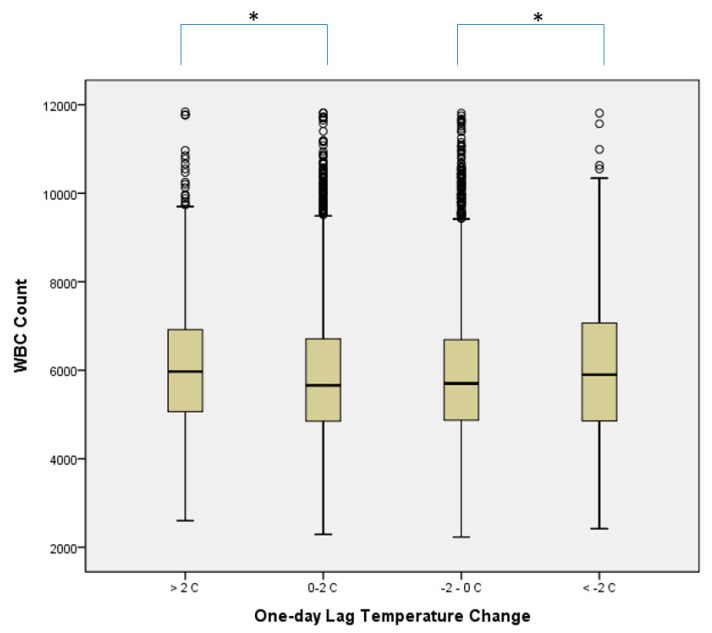
Box and Whisker plots of WBC counts by the difference values of one-day lag temperature change; * *p* < 0.05 by Scheffe test.

**Figure 3 ijerph-18-06971-f003:**
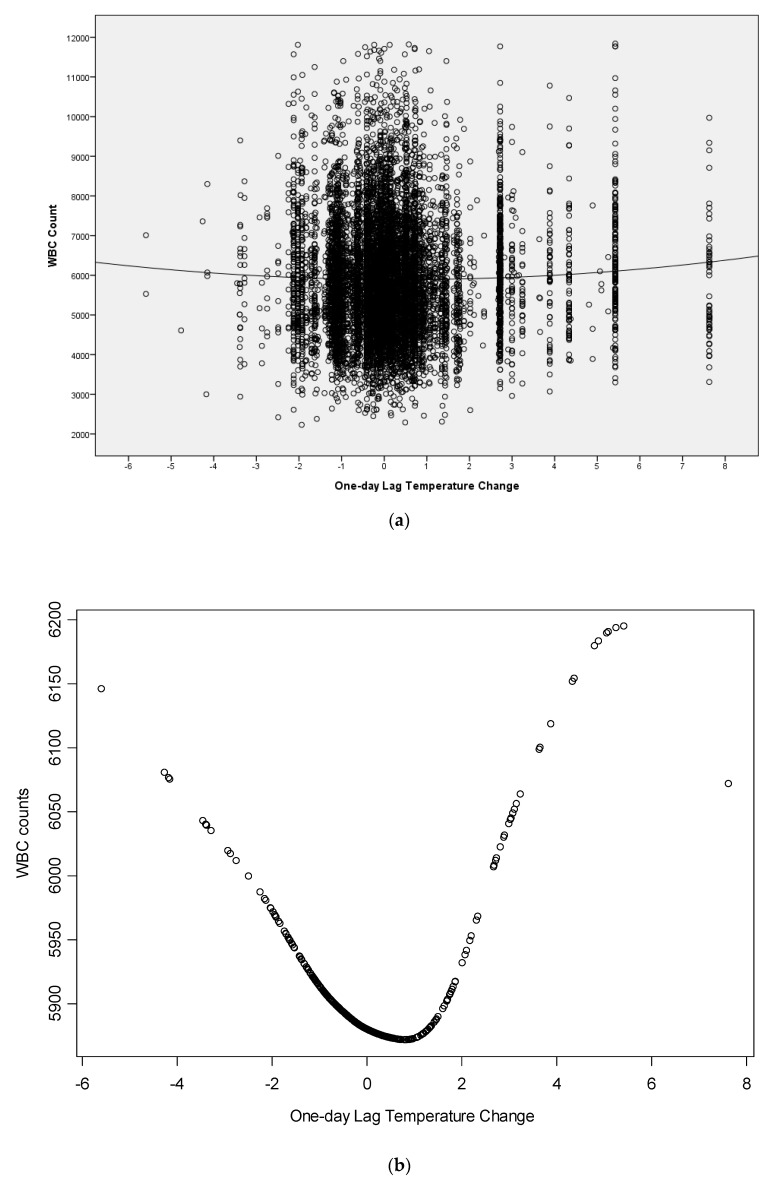
(**a**) Scatter plot of total WBC counts by one-day lag temperature with trend line of generalized additive model (GAM) procedure. The *p*-value of the GAM analysis is 0.000154; (**b**) zoom in the trend line at the central area.

**Figure 4 ijerph-18-06971-f004:**
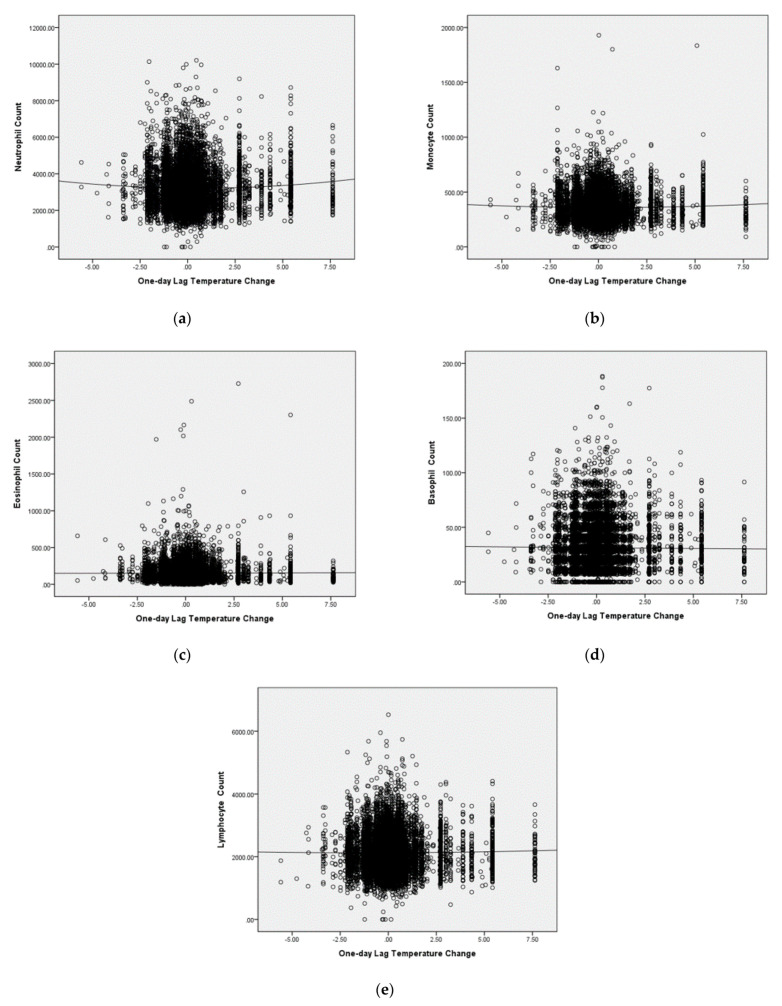
The scatter plot graphs of counts of WBC differential types and trend lines of GAM: (**a**) neutrophil, the *p*-value of GAM analysis is 0.000219; (**b**) monocyte, the *p* value of GAM analysis is 0.00000157; (**c**) eosinophil, the *p* value of GAM analysis is 0.000214; (**d**) basophile, the *p* value of GAM analysis is 0.0167; (**e**) lymphocyte, the *p* value of GAM analysis is 0.305.

**Table 1 ijerph-18-06971-t001:** Demographic characteristics of participants.

Variables			Total	ΔT ^1^ > 2 °C	0°C ≤ ΔT ≤ 2 °C	−2 °C ≤ ΔT < 0 °C	ΔT < −2 °C
**Days**			331	32	134	147	18
**N**			9278	891	3908	4168	311
**Age, years (mean ± SD)**			54.3 ± 5.9	53.8 ± 5.7	54.2 ± 5.7	54.4 ± 5.9	55.0 ± 6.1
**Gender, male**			4200 (45.3%)	340 (38.2%)	1661 (42.5%)	1836 (44.1%)	142 (45.7%)
**BMI, kg/m^2^ (mean ± SD)**			24.7 ± 3.8	25.0 ± 3.8	24.6 ± 3.4	24.8 ± 4.1	25.2 ± 3.6
**Education**	College/Graduate		1259 (13.0%)	108 (12.2%)	506 (13.0%)	528 (12.8%)	30 (9.8%)
	High school		3874 (40.0%)	371 (42.0%)	1587 (40.8%)	1626 (39.4%)	108 (35.2%)
	Elementary school		3328 (34.3%)	293 (33.1%)	1326 (34.1%)	1433 (34.7%)	119 (38.8%)
	Illiterate		1228 (12.7%)	112 (12.7%)	468 (12.0%)	540 (13.1%)	50 (16.3%)
**Cigarette smoking**		current	1457 (15.7%)	113 (13.0%)	558 (14.5%)	652 (15.9%)	60 (19.5%)
		former	501 (5.2%)	46 (5.3%)	189 (4.9%)	222 (5.4%)	17 (5.5%)
		never	7628 (79.6%)	708 (81.7%)	3106 (80.6%)	3324 (78.7%)	231 (75%)
**Alcohol consumption**			1383 (14.3%)	89 (10.1%)	591 (15.3%)	603 (14.6%)	44 (14.3%)
**Betel quid use**			273 (2.8%)	24 (2.7%)	102 (2.6%)	126 (3.1%)	8 (2.6%)

^1^ ΔT, difference of one-day lag apparent temperature.

**Table 2 ijerph-18-06971-t002:** Counts and percentage of WBCs by different value of ΔT.

Variables	Total	ΔT ^1^ > 2 °C	0 °C ≤ ΔT ≤ 2 °C	−2 °C ≤ ΔT < 0 °C	ΔT < −2 °C	*p* ^2^
**Days**	331	32	134	147	18	
**N**	9278	891	3908	4168	311	
**Counts of**						
Total WBC	5910.7 ± 1477.3	6110.1 ± 1460.3	5876.4 ± 1459.8	5889.9 ± 1478.2	6064.8 ± 1593.1	<0.0001
Neutrocyte	3232.2 ± 1129.3	3389.6 ± 1143.8	3201.8 ± 1107.6	3221.3 ± 1137.0	3314.4 ± 1221.9	<0.001
Monocyte	360.4 ± 128.4	371.8 ± 128.6	357.3 ± 126.3	358.0 ± 124.0	389.9 ± 172.9	<0.001
Eosinophil	142.1 ± 130.7	157.4 ± 166.4	153.5 ± 166.4	149.4 ± 129.1	163.3 ± 119.5	0.116
Basophil	30.9 ± 20.2	30.8 ± 20.8	31.8 ± 21.1	30.7 ± 21.0	33.75 ± 29.2	0.05
Lymphocyte	2130.0 ± 617.4	2161.8 ± 583.1	2130.0 ± 618.3	2120.6 ± 623.5	2163.9 ± 619.3	0.234
**Percentage of**						
neutrocyte (%)	53.99 ± 9.02	54.78 ± 8.76	53.81 ± 8.96	54.00 ± 9.25	53.87 ± 8.45	0.041
monocyte (%)	6.15 ± 1.77	6.13 ± 1.69	6.15 ± 1.87	6.14 ± 1.64	6.45 ± 2.37	0.029
eosinophil (%)	2.57 ± 1.92	2.55 ± 2.18	2.62 ± 1.92	2.53 ± 1.89	2.72 ± 1.83	0.118
basophil (%)	0.54 ± 0.35	0.51 ± 0.33	0.55 ± 0.35	0.53 ± 0.35	0.56 ± 0.37	0.005
lymphocyte (%)	36.66 ± 8.38	36.05 ± 8.06	36.84 ± 8.32	36.63 ± 8.58	36.40 ± 7.89	0.074

^1^ ΔT, difference of one-day lag apparent temperature. ^2^ ANOVA test (analysis of variance) comparing the groups of different ΔT.

## Data Availability

Data was obtained from research project (number NSC 99-2632-B-037-001-MY3) and are available with the permission of Ministry of Science and Technology.

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
