# Peer review of "The Association Pattern between Ambient Temperature Change and Leukocyte Counts"

_ijerph, 2021, doi:10.3390/ijerph18136971_

Round 1

Reviewer 1 Report

The article entitled "The pattern of association between changes in ambient temperature and leukocyte counts" examined the association between changes in ambient air temperatures and peripheral leukocyte counts in Kaohsiung, Taiwan.

The research is relevant and of great impact for the study of the influence of the environment on health. However, I would like to make a suggestion for better robustness and solidity to the results so that the work is published in the IJRPH.

The methodology is adequate, but I wonder if it would not be more appropriate to use some thermal comfort index based on the thermal sensation and sensitivity in relation to thermo-physiological parameters, such as the Equivalent Physiological Temperature (PET). This index has already been calibrated for the study region and can be calculated from data on temperature, humidity, wind and solar radiation. PET results may be more consistent with what is intended in this study.

Reviewer 2 Report

The authors examine the impact of meteorological variables, especially day-to-day changes in apparent temperature, on various human health outcomes.  While the findings are interesting, more detail is needed throughout much of the manuscript.

The language is in need of major revision; I recommend getting a native English speaker to examine the manuscript prior to resubmission.

Abstract:

Not clear why climate change has any impact here.

Stable weather conditions often mean something very different than what the authors are describing.

Introduction:

Again, I am unclear why climate change is mentioned.  This is a two-year study examining short-term changes in temperature.  Has a long-term warming been impacting the number of short-term temperature changes that the authors are examining here?  If not, I suggest the authors remove all discussion of climate change since it strikes me as irrelevant.

The introduction is rather brief, and it would be valuable for the authors to explore more specifically how rapidly-changing short-term temperatures can impact human health.

Materials & Methods:

Please provide more geographic information regarding where this study was conducted.

Please provide more information regarding the meteorological variables.  At what time of day were the data collected?  If the temperature and dew point data are daily averages or daily max/min values, then the apparent temperature values are likely invalid.

By “air quality data”, I assume the authors are referring to meteorological data and not pollution?

To summarize, the materials and methods section needs to be expanded considerably.

Reviewer 3 Report

Int. J. Environ. Res. Public Health

Ms. Ref. No.: ijerph-1244707

Title: Association pattern between ambient temperature change and leukocyte counts

The presented work deals with outdoor temperature change and its impact on human health, using white blood cells (WBC) counts as disease marker, in Kaohsiung city as a case study. Undoubtedly this is an important issue. This is an original research, where the authors used generalized additive models and found V-shaped pattern association between WBC counts and temperature change, where two-degree Celsius temperature changes with one-day lag was statistically significant. A series of major comments are provided here that support the revision of the manuscript that will strengthen the scientific soundness and make it appropriate for publishing.

Major comments:

R27, R42-43 and elsewhere below (R58-59, RR61-63, RR94-95, R140, R159, and etc.): It is not obvious, if you are looking at temperature change general (in timing perspective), or temperature change between two consecutive days (that is temperature difference with the previous day, or short-term temperature variability), and its impact on human health. Please divide these two issues.

Please add the DETAILED description of study area climate in Materials and Methods section, with Koppen-Geiger classification, etc.

RR97-98: why do you know that association between WBC counts and temperature changes (variability?!!!) have a U- or V-shaped pattern, please give a proof.

Discussion. Please add something like recommendations: why your findings are important, how health practitioners and physicians, or patients can use your results, etc.

Minor comments:

R43: What does ‘disease burden’ mean, is it something different with morbidity or mortality?

R45: You can add paper by Yi Jiang et al. (2020) Development of a health data-driven model for a thermal comfort study, https://doi.org/10.1016/j.buildenv.2020.106874, to reference list for optimum temperature of minimum mortality.

RR51-52: What does ‘incidence, morbidity and mortality of diseases’ mean? Probably, should be ‘incidence of diseases, morbidity and mortality’?!

R59: Please add ‘variations’: ‘mechanism of how ambient temperature VARIATIONS influence the human health’.

R81: please add ‘daily’ air temperature…(if it was daily data – not obvious from your wording here).

RR81-82: why you need data on total cloud cover, wind speed and wind direction, if only daily temperature and dew point temperature are used for Tapp.

RR84-85 Please explain why you use Tapp but not any other biometeorological index? Why Tapp THI is better than >165 human thermal climate indices which are proposed by bioclimatological community, e.g., THI, or PET or UTCI, or any other?

R87: ‘Air quality’ term is normally used for air pollution. Please refer here to ‘Weather data’.

R104-105 and equation (3): tobacco? smoking? alcohol? Please specify those terms in both equation and description.

RR114-115 (and fig.1): Please check if you use weather data from air-quality monitoring station, or from meteorological station (R79).

R119: add (kg) to the weight measure.

R122: is this ‘one-day lag temperature’ or one-day lag in ‘difference of apparent temperature between two days’? Please check if you use T or Tapp here and elsewhere below (Tables 1 and 2, etc.?

Tables 1 and 2: please use Tapp> 2°C (instead of 2°C < Tapp).

R162: Please use ‘the higher apparent temperature difference between two consecutive days’ instead ‘The more of temperature change’

R166: your research findings are about V-shaped pattern association between apparent temperature VARIATION’ and WBC count’ but not ‘temperature’, this is not about optimum temperature of minimum mortality.

Reviewer 4 Report

I appreciate the scientific quality of this draft which fits the publication criteria.

The introduction, the study methodology, the results, the discussions and the conclusions are well presented, consistent and clear. As such, this study seems to me suitable for publication.

Related to the temperature change,could the authors consider in their analysis or in the discussions , the speed of change and the midnight temperature change? These may bring additional intresting information for further investigation on the impact of seasonal variation  or warming / cooling meteorological conditions.

Round 2

Reviewer 2 Report

Thank you to the authors for their hard work in attempting to make the requested changes.  Unfortunately, my main criticisms still hold.  To summarize:

  • The primary independent variable is invalid. Apparent temperature is an instantaneous index and cannot be calculated using average daily temperature and average daily dew point.  As an example, assume two measurements per day:  T=10, Td=10 and then T=20, Td=20.  The actual average AT using the authors’ equation would be 16.082.  However, using the authors’ methodology, it would be 15.6995, which is incorrect.  Both references cited by the author (42,43) use instantaneous measurements, and one reference even specifically highlights that researchers using this calculation for AT should use data from a fixed time of day.   
  • I still do not understand the continued focus on climate change. What does climate change have to do with day-to-day changes in AT?  Starting with the second sentence of the manuscript, the authors note:  “And the trend of global warming and climate changes have [sic] been getting more obviously [sic] in recent decades.  According to the synthesis report on climate change in 2014 by the Intergovernmental Panel on Climate Change (IPCC), the globally [sic] surface temperature (combined land and ocean) shows 0.85 [0.65 to 1.06] °C warmer [sic] in recent 3 decades [sic], from 1880-2012 [1].”  The authors provide no evidence that day-to-day changes in apparent temperature have fluctuated over time.  Increasing surface temperatures are irrelevant here since the authors aren’t examining it.
  • The English still needs substantial work (the two sentences above have five typos alone!), and I continue to recommend the authors use a native English speaker to proofread their manuscript.    

Reviewer 3 Report

Accept in present form

Author Response

Thank you, and appreciate for the reviewer’s suggestions.